# Alterations in the Epigenetic Machinery Associated with Prostate Cancer Health Disparities

**DOI:** 10.3390/cancers15133462

**Published:** 2023-07-01

**Authors:** Jenna Craddock, Jue Jiang, Sean M. Patrick, Shingai B. A. Mutambirwa, Phillip D. Stricker, M. S. Riana Bornman, Weerachai Jaratlerdsiri, Vanessa M. Hayes

**Affiliations:** 1School of Health Systems and Public Health, Faculty of Health Sciences, University of Pretoria, Pretoria 0084, South Africa; jenna.craddock@tuks.co.za (J.C.);; 2Ancestry and Health Genomics Laboratory, Charles Perkins Centre, School of Medical Sciences, Faculty of Medicine and Health, University of Sydney, Camperdown, NSW 2006, Australia; 3Department of Urology, Sefako Makgatho Health Science University, Dr George Mukhari Academic Hospital, Medunsa 0208, South Africa; 4Department of Urology, St Vincent’s Hospital, Darlinghurst, NSW 2010, Australia; 5Manchester Cancer Research Centre, University of Manchester, Manchester M20 4GJ, UK

**Keywords:** prostate cancer, somatic alteration, epigenomics, epigenetic machinery, African ancestry, southern Africa, health disparity

## Abstract

**Simple Summary:**

African ancestry is a significant risk factor for aggressive prostate cancer (PCa), with southern African ethnicity conferring a nearly 3-fold increased global risk for associated mortality. It is well understood that epigenetic alterations drive PCa initiation and progression, coupled with somatic alterations in genes encoding epigenetic enzymes. However, differences in the somatic alterations in these genes in African- versus European-derived prostate tumors and how they may contribute to PCa health disparities has yet to be investigated, which forms the objective of this study. With current PCa care almost exclusively based on and tailored for men of European ancestry, the identification of African-specific novel PCa epigenetic cancer drivers (*n* = 18), including therapeutic potential (6/18), offers clinical significance with the possibility of improving healthcare approaches and health outcomes for men of African ancestry.

**Abstract:**

Prostate cancer is driven by acquired genetic alterations, including those impacting the epigenetic machinery. With African ancestry as a significant risk factor for aggressive disease, we hypothesize that dysregulation among the roughly 656 epigenetic genes may contribute to prostate cancer health disparities. Investigating prostate tumor genomic data from 109 men of southern African and 56 men of European Australian ancestry, we found that African-derived tumors present with a longer tail of epigenetic driver gene candidates (72 versus 10). Biased towards African-specific drivers (63 versus 9 shared), many are novel to prostate cancer (18/63), including several putative therapeutic targets (*CHD7*, *DPF3*, *POLR1B*, *SETD1B*, *UBTF*, and *VPS72*). Through clustering of all variant types and copy number alterations, we describe two epigenetic PCa taxonomies capable of differentiating patients by ancestry and predicted clinical outcomes. We identified the top genes in African- and European-derived tumors representing a multifunctional “generic machinery”, the alteration of which may be instrumental in epigenetic dysregulation and prostate tumorigenesis. In conclusion, numerous somatic alterations in the epigenetic machinery drive prostate carcinogenesis, but African-derived tumors appear to achieve this state with greater diversity among such alterations. The greater novelty observed in African-derived tumors illustrates the significant clinical benefit to be derived from a much needed African-tailored approach to prostate cancer healthcare aimed at reducing prostate cancer health disparities.

## 1. Introduction

The epigenetic machinery comprises genes that encode proteins involved in regulating chromatin organization, histone modifications, DNA methylation, non-coding RNA, and RNA methylation [1,2]. Many of these genes are directly or indirectly linked to the epigenetic regulation of gene expression. The components of the epigenetic machinery can establish and maintain the epigenetic programming of a cell, or they can dynamically alter it, thereby affecting both the identity and function of cells. Either way, proteins of the epigenetic machinery collectively interact with complex interdependence and interactivity. Cancer genome sequencing has increased our understanding of epigenetic dysregulation as a feature of tumor development. Independent of genomic aberrations, prostate tumors display DNA methylation patterns that differ from those of normal tissue. For example, the tumor suppressor *GSTP1* promoter region is typically hypermethylated in prostate cancer (PCa), resulting in the loss of its expression [3,4]. Androgen stimulation in PCa has the capacity to recruit histone modifiers, triggering changes in the chromatin states of PCa cells [5,6]. Ultimately, these epigenetic changes confer a more active or inactive chromatin state, dysregulating gene expression, and thereby causing the downregulation or silencing of tumor suppressor genes or a loss of regulation of genes that promote carcinogenesis. Although aberrant DNA methylation and disordered chromatin organization have long been recognized as features of cancer, the exact mechanisms driving epigenetic dysregulation are only just beginning to be understood.

A number of studies have shown that driver gene mutations in several cancer types are enriched for epigenetic machinery genes, including PCa. While most individual genes are mutated infrequently, epigenetic machinery genes, as a class, are some of the most frequently mutated in PCa [7,8]. However, there are a number of epigenetic modulators revealed to carry frequent and recurrent mutations. In PCa, variants of this nature have been identified in mediators of DNA methylation (e.g., *TET2*, *MBD1*), histone acetylation (e.g., *KAT6B*, *ARID4A*), histone methylation (e.g., *KMT2C*, *SETD2*), as well as in chromatin remodelers (e.g., *ARID1A*, *SMARCA1*) [7,8,9,10,11]. The putative driver mutations are often truncating or missense mutations [8], giving rise to non-functional proteins or proteins with altered functionality. In addition to small somatic variants, prostate tumors are prone to acquire more complex variation, including structural variations (SVs) and copy number (CN) aberrations [12]. Of relevance to epigenetic machinery are double-stranded breaks, as well as the alteration of CpG methylation, local histone methylation, and the chromatin structure at the DNA repair sites, with consequent altered gene expression [13,14,15]. Zhang et al., 2019 [16], showed overall SV burden to be associated with global hypomethylation and increased expression of methyltransferase genes across cancer types. Epigenetic dysregulation by imbalanced genomic rearrangements has also been demonstrated in tumors, with the consequences of such rearrangements often being local, in that CN changes in a gene will alter the DNA methylation or gene expression of nearby genomic regions [17].

What remains to be considered is the potential contribution of somatic alterations within the epigenetic machinery and its relevance to PCa health disparities. Notably, genetic ancestry is a significant risk factor for aggressive PCa, specifically African ancestry. Within the United States, African American men are 1.7 times more likely to be diagnosed with, and over twice as likely to die from, PCa than European ancestral American men, reaching 3.1-fold greater incidence for men younger than 65 years at diagnosis [18]. Globally, mortality rates are 2.7-fold greater for men from Sub-Saharan Africa [19]. While both genetic (common and rare variants) [20,21] and non-genetic (socioeconomic and cultural) contributing factors have been proposed [22], studies focused within populations from Sub-Saharan Africa have been scarce [23,24]. Conversely, ancestral differences in epigenetic aberrations have been observed for PCa, including genome-wide aberrant methylation patterns [25,26,27,28]. Gene-specific examples include the hypermethylation of *CD44* in prostate tumors derived from African-American compared to European ancestral Americans, which was positively correlated with tumor grade [29], and the hypermethylation of *RARB*, which was significantly associated with a higher risk of PCa in African-American over European ancestral American men [30]. While genomic aberrations in epigenetic machinery components have been studied previously for PCa [7,8,11], the contribution to ancestral differences associated with health outcomes is yet to be investigated, specifically within the context of Sub-Saharan Africa. Consultation of the PathCards database [31] and a review of the literature [1,7,8,32,33,34] identified 656 genes (in)directly related to all known epigenetic processes, several of which have been implicated previously in PCa. Using a unique resource of prostate tumor genomic data derived from 113 men of Southern African and 53 men of European Australian ancestry [35], we set out to determine whether genomic aberrations in these epigenetic machinery components could, at least in part, explain the ancestral disparity observed for PCa.

## 2. Materials and Methods

### 2.1. Patient Clinical Characteristics and Genomic Data

Patients were recruited as part of the Southern African Prostate Cancer Study (SAPCS), with approval granted by the University of Pretoria Faculty of Health Sciences Research Ethics Committee (with US Federalwide Assurance FWA00002567 and IRB00002235 IORG0001762) in South Africa (43/2010). In Australia, participant recruitment was approved by the St. Vincent’s Human Research Ethics Committee (HREC) (SVH/12/231), with genomic data generation approved by the St. Vincent’s HREC (SVH/15/227). Additional study-specific approval was granted by the University of Pretoria Faculty of Health Sciences Research Ethics Committee (504/2022). Fresh blood-tumor paired deep sequenced whole genome data were generated, as previously published [35]. In brief, data were generated using 2 × 150 cycle paired-end Illumina HiSeq/NovaSeq sequencing, with reads aligned to the GRCh38 reference genome with alternative contigs, achieving a mean depth of coverage of 90X (range 28–139X) for tumors and 46X (range 30–97X) for blood. Both germline and somatic variants were called as previously described [35], including single nucleotide variants (SNVs) and small (<50 bases) insertions and deletions (indels), structural variants (SVs, >50 bases), and somatic copy number alterations (CNAs). While somatic variant frequencies and CNAs were used to determine tumor purities, which ranged from 13% to 88%, germline 7,472,833 biallelic SNVs were used to determine patient genetic ancestral fractions using fastSTRUCTURE v.1.0 population sub-structure analyses [36]. From the 183 patients included in the Jaratlerdsiri et al. 2022 study [35], patients were excluded if they lacked a positive PCa diagnosis (*n* = 6), were Brazilian (*n* = 7), were of admixed ancestry (defined as <85% genetic contribution from a single ancestral identifier, *n* = 3), or if their tumors were hypermutated (defined as >30 mutations per Mb, *n* = 2). Of a total of 165 treatment-naïve PCa patients included in our study, 4/109 (3.7%) of the African and 3/56 (5.4%) of the European ancestral patients lacked any genomic aberrations within the epigenetic machinery, leaving 105 and 53 for further interrogation, respectively. Both cohorts were biased towards aggressive disease, representing the International Society of Urological Pathology (ISUP) Group Grading 4 and 5 in 73% of African and 85% of European-derived tumors, respectively (see Appendix A for a summary of clinical characteristics).

### 2.2. Epigenetic Process Group Classification

Using the PathCards database to identify genes that map the epigenetic process pathways [31], we identified 656 epigenetic process-related genes. A SuperPath represents a cluster of one or several pathways that are grouped together based on the similarity of their associated genes (Appendix A). Based on a review of the literature, conducted in July 2022, additional epigenetic process-related genes were included due to frequent reference and/or previous mention of their relationship to PCa [1,7,8,32,33,34]. As several epigenetic processes regulate the chromatin state, we further subdivided the 656 genes into their Epigenetic Process Group (EPG). EPG 1 genes are involved in chromatin organization and regulation (*n* = 530 genes, Appendix A), EPG 2 genes in histone modifications (*n* = 240, Appendix A), EPG 3 genes in DNA methylation (*n* = 101, Appendix A), EPG 4 genes in RNA regulation (*n* = 136, Appendix A), and EPG 5 genes in the epigenetic regulation of gene expression (*n* = 253, Appendix A). Due to the multifunctional nature of these genes, a number have been assigned to multiple EPGs, while others are exclusive to a single EPG.

### 2.3. Tumor Mutational Burden (TMB), Damaging Variant Detection, and Mutational Frequency Analysis

Whole genome tumor mutational burden (TMB) was calculated for each patient by taking the total number of small somatic variants (SNVs and indels), divided by the total genome size (3088 Mbp). For each EPG classification, we defined mutational burden as the total number of small somatic coding variants present in a respective collection of epigenetic machinery genes, divided by the total coding size (Mbp) of that gene collection. The damaging variant mutational burden was defined as the total number of potentially damaging variants present in a respective collection of epigenetic machinery genes, as per functional impact prediction, divided by the total coding size (Mbp) of that gene collection. Coding the genome size for each EPG was mined using the Ensembl v.108 BioMart online data retrieval tool [37,38].

For each EPG, two approaches were used to identify potentially damaging variants and genes, including functional impact prediction and mutational recurrence. Specifically, the SIFT [39] and PolyPhen [40] scores for epigenetic process coding gene variants were determined using the SNPnexus v.4 annotation tool [41]. A variant was considered to be potentially damaging if identified by SIFT as “Deleterious” or “Deleterious–Low Confidence”, or if identified by PolyPhen as “Possibly Damaging” or “Probably Damaging”.

For recurrently mutated genes, we applied the computational tool DrGaP (driver genes and pathways) [42] to synonymous and non-synonymous somatic variants to determine the probability of each variant occurring by chance. DrGaP defines driver genes as those for which the non-synonymous mutation rate is significantly higher than the background mutation rate (BMR), while integrating biological variables such as the length of protein-coding regions. Using DrGaP, we defined significantly altered genes as those with a false discovery rate (FDR) < 5%, using the Benjamini-Hochberg (BH) method.

Of the potentially damaging variants and genes found, we further identified genes that overlap with the Pan Cancer Analysis of Whole Genomes (PCAWG) compendium of mutational drivers [43]. To visualize patients’ overall somatic variant landscape, we used the maftools R package [44] to generate summary oncoplots for each of the EPGs.

### 2.4. Integrative Analysis of Epigenetic Machinery-Driven Prostate Cancer Subtypes

We performed integrative clustering of three genomic data types (small somatic variants, SVs, and somatic CNAs) overlapping epigenetic process-related genes for 158 patients using the MOVICS (Multi-Omics integration and VIsualization in Cancer Subtyping) R package [45]. We ran the optimal cluster number identification function, with clusters ranging from 2 to 8, and MOVICS arbitrarily assigning an optimal cluster number of 8 for the variant data. The Cluster Prediction Index (CPI) and Gap statistic encouraged consideration of a cluster number of 3 as optimal (Appendix A). We executed ten classical clustering algorithms to subtype patients with different molecular features (Appendix A), from which the resulting consensus matrix and silhouette plot ultimately demonstrated an optimal cluster number of 3 (Appendix A), with which we proceeded for our analyses. Our prior research also informed this decision, where through whole tumor genome interrogation, we identified four PCa taxonomies, termed Global Mutational Subtypes (GMSs), which differentiated patients by their ancestries [35]. We then visualized the consensus clustering result with a heatmap and annotated top features based on the posterior probability for each genomic feature as a driver, estimated by iClusterBayes [46,47,48], while additional hierarchical clustering was performed for CNAs [49]. The relationship between cancer subtypes and various clinical features was assessed.

### 2.5. Statistical Analyses

For continuous variables, group means were compared using the Mann–Whitney U test, while for categorical variables, the groups were compared using Fisher’s exact test. A *p*-value < 0.05 was regarded as statistically significant. Biochemical relapse (BCR) survival probability and cancer survival probability for European patients were analyzed with the Kaplan–Meier method, followed by group comparison for significance using a log-rank test.

## 3. Results

### 3.1. Tumor Mutational Landscape

We have previously shown that African-derived prostate tumors present with an elevated TMB and potentially damaging somatic variants [35,50]. When considering the epigenetic machinery genes, 73.3% of African-derived tumors and 62.3% of European-derived tumors harbored altered genes, representing an average of 1.8 (range 1 to 20) and 1.4 (range 1 to 3) damaging variants, respectively. Compared to patient-matched whole genome data [35], we found the same trend (although not significant) when considering epigenetic machinery-restricted mutational burden (Appendix A and Appendix A) and damaging variants (Table 1 and Appendix A, Appendix A) for all genes and EPGs 1, 2, 4, and 5. Notably, each of the five EPGs displayed a higher mean mutational burden than expected when compared to the mean TMB observed for the whole genome, irrespective of patient ancestry. In contrast, EPG 3 showed the highest and equal mean mutational burdens (Appendix A) and number of damaging variants (Appendix A) for both Africans and Europeans (Appendix A).

To substantiate the potential carcinogenic nature of the identified damaging genes (Table 2 and Table 3), we correlated for recurrence using the PCAWG compendium of mutational drivers [43]. Interestingly, *KMT2C*, identified here as a potentially damaging gene and by the PCAWG as a candidate cancer driver, although not predicted to contain functionally impactful variants, showed the highest recurrence rate, irrespective of ancestry (7.1% European, 5.5% African). The following most recurrent genes in African-derived tumors were *CHD3* and *ARID1B* (4.6% and 3.7%, respectively), and in Europeans, were *BRD7*, *KDM6A*, *KDM6B*, *KMT2A*, *KMT2B*, and *RANBP2* (each 3.6%). Seven genes presented in tumors from three or more patients of African ancestry, while being notably absent from European-derived tumors and included *CHD3* (5 patients), *ARID1B* (4), *HDAC4* (3), *ARID1A* (3), *CHD1* (3), *PRDM16* (3), and *STAG2* (3). Conversely, genes exclusively altered in two or more European-derived tumors included *KMT2B* (2 patients) and *RANBP2* (2). In each EPG, African-derived tumors displayed a greater mutation frequency for most genes (likely reflective of a greater sample number), along with richer diversity in variant types than European-derived tumors (Figure 1). In terms of mutation frequency, the top gene in EPG 1 and EPG 2 is *KMT2C*; in EPG 3, the top genes are *CBX2*, *DNMT3B*, and *TDG*; and in EPG 4 and EPG, the top gene is *CHD3*.

### 3.2. Integrative Clustering Analysis

Hierarchical consensus clustering through joint analysis of somatic variant data (small variants, SVs, and CNAs) for epigenetic machinery genes identified three PCa subtypes (Epigenetic Cancer Subtypes 1–3, i.e., ECS1-ECS3). These three subtypes are presented for 158 patients, as shown in Figure 2A (columns). Among genes with significantly different alteration frequencies between the cancer subtypes, ECS1 demonstrated a higher frequency of small somatic alterations and SVs compared to ECS2 and ECS3 (Appendix A), although all three ECSs were rather mutationally quiet. Overall, SV demonstrated similarity between the subtypes, while CNAs appeared to be the strongest indicator for subtype clustering. While ECS1 showed both CN gains (median 1 gene gained per tumor, range 0 to 38 genes gained) and losses (median 0 genes deleted per tumor, range 0 to 8 genes deleted), ECS2 and ECS3 were characterized by substantial CN gains (median 9 genes gained per tumor, range 0 to 187 genes gained) and CN losses (median 5.5 genes deleted per tumor, range 0 to 23 genes deleted), respectively, and were both African-predominant (91% and 67% African, respectively; *p* < 0.001; Appendix A). The four European patients allocated to ECS2 resided within Australia (*n* = 3) or South Africa (*n* = 1).

Feature selection identifies genomic features that make important contributions the oncogenic processes and drive the integrative clustering. Using iClusterBayes, we calculated the posterior probability for each genomic feature as a driver. We found the top five identified features were dominated by genes belonging to EPG 1, followed by EPG 2 and EPG 5. Posterior probability was only high for small somatic mutation-identified features, with 30 potential drivers identified in total (posterior probability > 0.5, Appendix A), a number of which were in agreement with identified damaging genes (Table 2 and Table 3). Irrespective of ancestry, potential drivers and damaging variant-containing genes included *SETD1B*, *CHD4*, and *BRD4*, while we observed a longer tail of unique drivers within Africans, including *SRCAP*, *ARID1A*, *HCFC1*, *BRD1*, *POLR1B*, *STAG2*, *VPS72*, *UBTF*, *MXD1*, *KDM6B*, *HDAC4*, *ELL*, *RANGAP1*, *SMARCA4*, *KDM2B*, and *NCOR1*. This suggests that different epigenetic mechanisms, particularly chromatin organization and regulation, play a role in PCa among Africans.

Through whole-genome PCa molecular taxonomy of the same sample source, all patients were classified into one of four recently described GMSs [35]. In brief, both GMS-A and GMS-C are ethnically diverse, marked by a mutationally quiet landscape and substantial CN losses, respectively. Conversely, GMS-B and GMS-D are African-predominant, with GMS-B demonstrating substantial CN gains, and GMS-D presenting a mutationally noisy landscape, including CN gains and losses. We found significant correlation between our ECSs and GMSs (*p* < 0.001, Appendix A). Of the ECS1 tumors, 94% had previously clustered with GMS-A, while the ECS2 tumors were dominated by GMS-B (47%), mapping exclusively to African-specific ECS2 tumors, and GMS-A (44%). In contrast, all European-derived ECS2 tumors belong to GMS-A. ECS3, characterized by near-equal contributions from GMS-A (50%) and GMS-C (48%), demonstrated the African-European GMS-C almost exclusively. Finally, the African-specific GMS-D tumors were the least represented of the subtypes, showing no favorable clustering with distribution across ECS1 to ECS3.

As CNAs appear to be the strongest determinant for ECS clustering, we performed hierarchical clustering on the CN data alone (Figure 2B). Recognizing three Epigenetic CN Cancer Subtypes (EcnCS, Appendix A), we found EcnCS2 to be African-exclusive and dominated by GMS-B (91%), while EcnCS3 was European-dominant (82%), representing only GMS-C tumors. The ancestry diverse subtype EcnCS1 was predominated by GMS-A tumors (85%). Overall, the EcnCSs correlated significantly with our previously identified GMSs (*p* < 0.001) and ECSs (*p* < 0.001).

Through the availability of extensive follow-up data for our European Australian patients (mean 127.4 months, range 37.4 to 214.3 months), we previously predicted significantly better clinical outcomes, defined as no biochemical relapse (BCR) and/or survival, for patients presenting with GMS-A over GMS-C tumors [35]. Here, we sought to correlate our identified ECSs and EcnCSs with clinical outcomes in our patient-matched European cohort. BCR-free probability revealed better clinical outcomes for ECS1 over ECS3 tumors, although not significant (Figure 3A, log-rank test, *p* = 0.15), and ECS2 tumors predicted poorer survival probability than ECS1 tumors (Figure 3B, log-rank test, *p* < 0.001). Notably, of the seven deaths recorded, three Australian European men presenting with ECS2 died within 172.5 months of their surgeries. Although not significant, when considering the CN subtypes, BCR-free probability showed a better clinical outcome for patients presenting with EcnCS1 over EcnCS3 (Figure 3C, log-rank test, *p* = 0.11), even when considering only the GMS-C tumors (Figure 3D, log-rank test, *p* = 0.32), which are characterized by poor clinical outcome.

## 4. Discussion

Overall, compared with European-derived tumors, African-derived prostate tumors presented with a higher burden of variants, as well as potentially damaging variants across epigenetic machinery genes. Although our findings were in line with our previous work, demonstrating a whole-genome African-elevated TMB [50], the epigenetic burden within Africans was not significantly higher than that in Europeans. When considering all epigenetic machinery genes, the African-derived tumors demonstrated a higher overall mutational frequency than the European-derived tumors, although this increase was not significant. In contrast to a recent genome-wide study, which found ~20% of prostate tumors harbored driver mutations across 12 epigenetic machinery genes [8], here we report frequencies of 52.3% for African- and 50.0% for European-derived tumors, which may be explained by a larger inclusivity of epigenetic regulators in our study (656 versus only 12 genes). Irrespective of patient ancestry, we found *KMT2C* to be the most frequently mutated PCa epigenetic regulator gene, concurring with the results of previous studies reporting frequencies of ~5–8% [7,8]. The type 2 histone lysine methyltransferase *KMT2C* is one component of chromatin remodeling machinery responsible for DNA promoter and enhancer regulation, ultimately promoting active chromatin conformations. With strong links to numerous cancer types, mutations in these components confirm their roles as tumor suppressors [51]. Irrespective of potentially damaging (Table 2) and recurrent (Table 3) driver gene classification, as was observed for the whole genome, African-derived tumors showed a longer tail of African-specific epigenetic gene candidates. In addition to *KMT2C*, the only recurrent driver genes to be shared between the ancestries are the well-known tumor suppressor genes *KDM6A* and *TP53*. In contrast to a lack of European-specific recurrent drivers, 35 African-specific recurrent driver genes were observed. The latter included putative loss-of-function PCa mutations previously reported for *ARID1A*, *ATRX*, *CHD1*, *CHD3*, *HDAC4*, *KMT2A*, *KMT2D*, *SETD2*, and *SMARCA1* [7], with *BRMS1*, *CARM1*, *EHMT2*, *GLI3*, *HDAC1*, *KDM6B*, *PRDM16*, *RBBP5*, and *REST* possessing known roles in PCa [52,53,54,55,56,57,58,59,60]. *ARID1B*, *ARID5B*, *BRWD1*, *EP300*, *HDAC3*, *NCOR2*, *PSIP1*, *SMARCA4*, *STAG2*, and *XPO1*, although reported by PCAWG [43], are new to PCa, leaving *CHD7*, *DPF3*, *ELP2*, *GATAD2B*, *NUP35*, *SETD1B*, and *TADA2B* as novel candidate drivers.

Taking a closer look at the epigenetic processes, in contrast to our whole genome data and EPGs 1, 2, 4, and 5 exhibiting an African ancestry-elevated burden, we consistently showed EPG3 alterations to be similar between the ancestries. Overall, this group of DNA methylation gene regulators appears to be highly conserved, as previously reported [61], with no recurrent drivers (Table 3) and potentially damaging variants in only two genes, *DNMT3B* and *TDG* (Table 2). *DNMT3B* is a DNA methyltransferase (DNMT) enzyme responsible for establishing and maintaining methylation in satellite sequences and gene bodies [62,63]. DNMT polymorphisms are associated with PCa progression by means of downregulatory tumor suppressor gene promoter methylation [64] and elevated *DNMT3B* expression in aggressive versus non-aggressive PCa cell lines [65]. Similarly, a damaging variant in *DNMT1* was observed in an African sample. *TDG*, or Thymine DNA Glycosylase, plays a key role in active DNA demethylation and in tumor suppression. Several polymorphisms in *TDG* are associated with increased risk for cancer, although this gene has also been found to act as an oncogene, promoting tumorigenesis [66,67,68]. It remains to be determined whether the *TDG* damaging variants identified in our study possess gain-of-function or loss-of-function properties. Ultimately, aberrant DNA methylation is a hallmark of cancer progression, and dysregulation of the DNA methylation machinery may lead to a reprogramming of the epigenomic landscape in cancer.

Using hierarchical consensus clustering for all somatic mutational types (small variants, SVs, and CNAs), we describe two epigenetic PCa taxonomies (ECS and EcnCS), which independently showed significant agreement with our previously-reported GMSs [35]. Showing extensive overlap among Europeans, both ECS3 and GMS-C tumors predicted a poorer clinical outcome over ECS1 and GMS-A tumors, respectively, demonstrating the bias of each GMS to an ECS. As such, our identified ECSs validate the whole genome-derived GMSs and are able to relatively distinguish those global subtypes based on only a subset of the genome, indicating a significant role for epigenetic mechanisms in PCa development. While numbers for recorded PCa-associated death are arguably small (7/50 Australians), it is notable that all three Australian European patients with tumors presenting with ECS2 succumbed to PCa, i.e., 42.9% of PCa deaths were associated with ECS2 tumors. Furthermore, as ECS2 is otherwise characterized by African predominance, specifically with ISUP group grading > 3 PCa (78% of African-derived ECS2 tumors versus 73% of all African-derived tumors), our data suggest that ECS2 is a predictor of poor outcome.

More aligned with the whole genome-derived GMSs [35], the EcnCSs showed ancestral distinction, including an African-specific subtype (EcnCS2), a European-predominant subtype (EcnCS3), and a shared subtype (EcnCS1). Notably, EcnCS2 defined by significant CN gain further defines ECS2, while almost exclusively incorporating all the African-specific GMS-B tumors (95.2%, 20/21). EcnCS3 further distinguished ECS3, and the poor outcome-associated GMS-C, as a singular cluster defined by epigenetic gene CN loss. Of the GMS-C tumors, EcnCS3 presented with a higher predominance of ISUP group grading 5 PCa (81.8%) over EcnCS1 (60.0%), with EcnCS3 predicting a poorer outcome for BCR than EcnCS1, indicating a more aggressive presentation for EcnCS3-GMS-C tumors. Suggesting that epigenetic CNAs alone have the potential to predict patient outcomes in our study, the relationship between CNAs and DNA methylation in cancer has been examined previously [17], although not at length. However, it is generally understood that a gene’s CNAs affect the DNA methylation of nearby genomic regions. These two processes may be negatively associated (i.e., DNA methylation decreases with copy number gain and vice versa), in which case, the effect is localized to CpG islands, or they may be positively associated, in which case, the open sea (genomic region beyond 4 kb from a CpG island border) is affected. Either way, it has been suggested that genome-wide DNA methylation changes in response to CNA events are likely initiated and maintained by some “generic” machinery. This is supported by the Sun et al. (2018) [17] finding that CNA events and their association with altered DNA methylation are similar across cancer types. Another observation common for several cancer types is the appearance of ancestral differences in DNA methylation patterns. This has been observed in PCa, in which African-American tumors display a higher prevalence of DNA hypermethylation at disease-related loci compared to European-American tumors [26]. Therefore, each of the epigenetic (copy number) cancer subtypes, with their distinct CNA events, likely give rise to distinct aberrant DNA methylation patterns. Whether those DNA methylation patterns cluster in agreement with the CN patterns is yet to be determined. Of course, aberrant DNA methylation does not arise only in response to CN gain/loss events. However, inclusion of patient-matched DNA methylation data could determine this.

As a function of hierarchical clustering, feature selection identified the top five genes for each variant type for potential driver gene classification. In rank order, based on posterior probability, the top five features for small somatic variant data were *RAI1*, *SETD1B*, *SRCAP*, *ARID1A,* and *MED26*; for SV data, they were *SMYD4*, *GATAD2B*, *PPARG*, *MEF2D,* and *SMARCAD1*; and for CNAs, they were *HMGA2*, *SMYD5*, *SUMO3*, *SP110,* and *RAG2*. Of the top five selected features for SV and CNA data, and for the 30 small somatic mutation-identified drivers (Appendix A), the genes that appear new to PCa, which are lacking in PCAWG and are African-specific, include *SP110*, *GATAD2B*, *RAI1*, *MED26*, *BRD1*, *POLR1B*, *VPS72*, *ELP5*, *UBTF*, *MXD1*, *DR1,* and *ELL*. Formerly considered to be a transcriptional regulator of circadian clock components in neuronal tissue, a recent study found *RAI1* to act as a tumor suppressor in esophageal cancer; prior to this finding, the functional role of *RAI1* in tumors was unknown [69]. *SETD1B*, an essential component of a histone methyltransferase complex, believed to have essential, even housekeeping, functions within cells [70], although playing no clear role in malignancy, has been reportedly mutated in gastric and colorectal cancers [71]. *MED26*, belonging to the Mediator complex (MED) gene family, while implicated in several cancer types, does not include PCa [72,73]. Additionally, a number of the epigenetic regulators specific to African tumors have been identified as potential therapeutic targets. Chromatin remodeler *CHD7*, a somatic driver candidate in colorectal cancer (CRC), promotes CRC cell growth by binding target gene promoters, encouraging an open chromatin conformation and subsequent transcription, whereas *CHD7* knockdown inhibits CRC cell growth [74]. Similarly, *POLR1B* knockdown induces lung cancer cell apoptosis [75], *VPS72* knockdown inhibits the proliferation, invasion, and migration of hepatocellular carcinoma (HCC) cells [76], and *UBTF* silencing suppresses melanoma cell proliferation [77]. *DPF3*, a chromatin remodeling cofactor significantly downregulated in breast cancer tissue, promoting the proliferation of breast cancer cells, has been suggested as a novel therapeutic target for breast cancer therapy [78]. Increased *SETD1B* expression in HCC positively correlated with tumor size, clinical stage, and liver cirrhosis. Decreased *SETD1B* expression was associated with increased patient survival times, identifying this histone methyltransferase as a potential therapeutic target in HCC [79]. While several of the African-specific drivers show clinical relevance, the remaining genes are not well-studied as therapeutic targets in cancer [80,81,82,83].

Ultimately, alterations to genes encoding epigenetic machinery components are increasingly recognized in many cancer types, including PCa. From this study, based on genes containing potentially damaging variants as per functional impact prediction and/or recurrence, as well as putative driver gene status, as defined by feature selection during hierarchical clustering, we have summarized the top genes in African and European-derived tumors (per EPG) that may be instrumental in epigenetic dysregulation and the subsequent development and/or progression of PCa (Figure 4). Identifying a number of putative drivers, *ARID1A*, *CHD4*, *HCFC1*, *STAG2*, *SMARCA4,* and *NCOR1* are known cancer driver genes [43]. Notably, there is extensive ancestral overlap among the top genes in all the EPGs. The assignment of numerous epigenetic machinery genes to more than one EPG is due to the multifunctional nature of these genes. For example, *CHD4*, or Chromodomain Helicase DNA Binding Protein 4, is the main component of the nucleosome remodeling and deacetylase (NuRD) complex that plays an important role in epigenetic transcriptional repression. *CHD4*/NuRD also regulates RNA synthesis [84]. As such, the multifunctionality of *CHD4* warrants its inclusion in EPGs 1, 2, 4, and 5. Rather than the top genes being epigenetic machinery components exclusive to a single EPG, this broad overlap is reminiscent of the previously discussed CN-DNA methylation aberration events, arising from some “generic machinery”, common in many cancer types. Indeed, epigenetic regulators are well-conserved and mutate infrequently. However, should epigenetic regulation be disrupted, as a class, perhaps the genomic alteration of a common core group of multifunctional epigenetic regulators will achieve this mutation, promoting tumorigenesis. Many of our top-identified genes have well-established roles across cancer types, further supporting the representation of these altered genes as a “generic machinery” promoting cancer. Yegnasubramanian describes alterations in epigenetic reprogramming to be almost universal in human cancers [7]. However, it is clear that African-derived tumors present with many more (ancestry-specific) possible cancer drivers than do European-derived tumors, highlighting the diversity by which epigenetic dysregulation and consequent tumorigenesis may arise in Africans.

## 5. Conclusions

Alterations to epigenetic machinery components dysregulate epigenetic programming, chromatin structure, and consequent transcription, a feature of PCa development and progression that is increasingly becoming better understood. Here, we describe somatic alterations within the epigenetic machinery and their relevance to PCa health disparities, with African-derived tumors demonstrating a longer tail of African-specific epigenetic driver gene candidates, a number of which are novel to PCa (*BRD1*, *DR1*, *ELL*, *ELP2*, *ELP5*, *GATAD2B*, *MED26*, *MXD1*, *NUP35*, *RAI1*, *SP110*, *TADA2B*) and some which are putative therapeutic targets (*CHD7*, *DPF3*, *POLR1B*, *SETD1B*, *UBTF*, *VPS72*). Here, we also described two epigenetic PCa taxonomies (ECS and EcnCS) that differentiate patients by ancestry, predict clinical outcomes, resemble whole-genome derived global subtypes, and identify more African-specific putative drivers, ultimately indicating a significant role for epigenetic mechanisms in PCa development. Identifying many more African-specific (versus European-specific) potentially novel PCa drivers highlights the urgency for African inclusion in precision medicine-informed healthcare approaches to ultimately reduce PCa health disparities and improve health outcomes for African men.

## Figures and Tables

**Figure 1 cancers-15-03462-f001:**
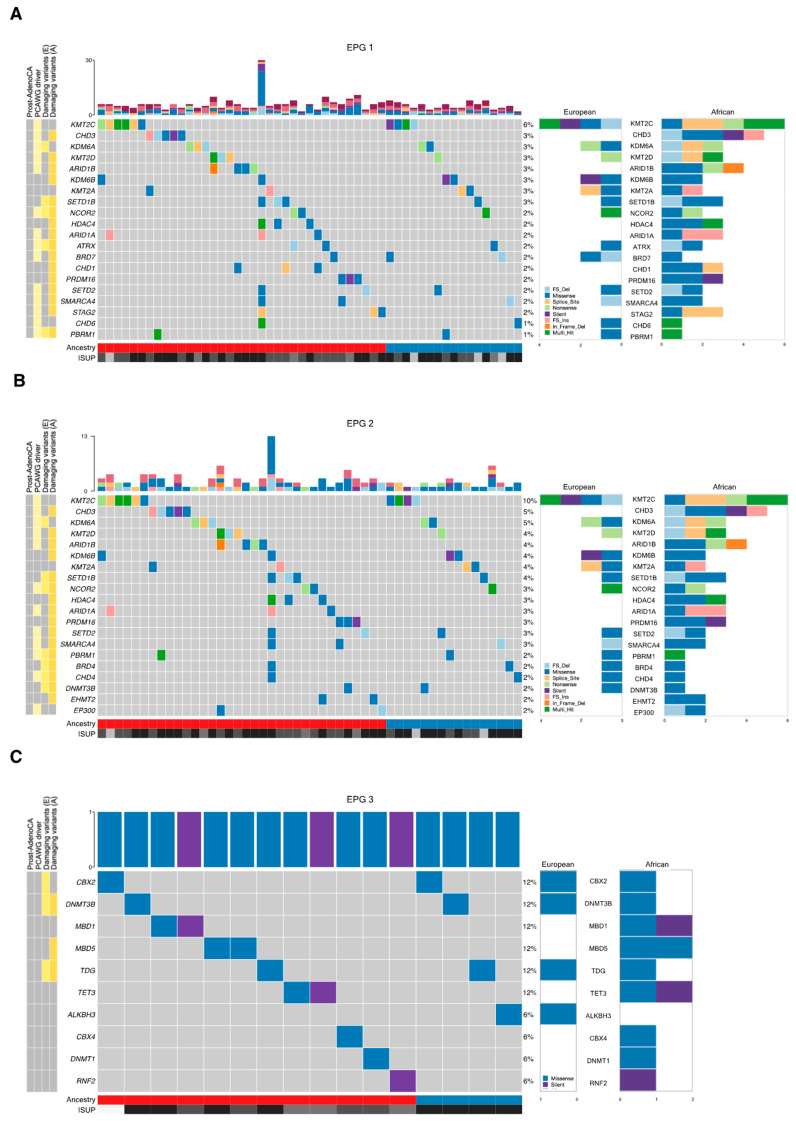
Somatic alteration landscape for each epigenetic process group. The top bar graph shows the number of non-synonymous variants and copy number alterations per tumor. The middle gene panel reports synonymous and non-synonymous variants in a maximum of 20 top altered genes. The bottom panels annotate sample ancestries and ISUP grades. The right-hand bar plots display European and African mutation frequencies, respectively. The left-hand panel indicates whether top genes identified in the oncoplot overlap with candidate cancer mutational drivers identified by the Pan Cancer Analysis of Whole Genomes and if so, whether the gene was identified as a candidate driver in prostate adenocarcinoma. Finally, the left-hand panel also indicates whether genes displayed in the oncoplot contain damaging variants, based on functional impact prediction, as described in Table 2. Yellow tiles indicate ‘yes’, grey tiles indicate ‘no’. Due to the hypermutated nature of these genes and their indirect epigenetic involvement in chromatin state regulation, the *TP53*, *SPOP*, and *FOXA1* genes were excluded from the oncoplots. (**A**) Epigenetic process group 1; (**B**) epigenetic process group 2; (**C**) epigenetic process group 3; (**D**) epigenetic process group 4; (**E**) epigenetic process group 5. A, African; E, European; FS_Del, frameshift deletion; FS_Ins, frameshift insertion; In_Frame_Del, in-frame deletion; ISUP, International Society of Urologic Pathologists; PCAWG, Pan Cancer Analysis of Whole Genomes; Prost-AdenoCA, prostate adenocarcinoma.

**Figure 2 cancers-15-03462-f002:**
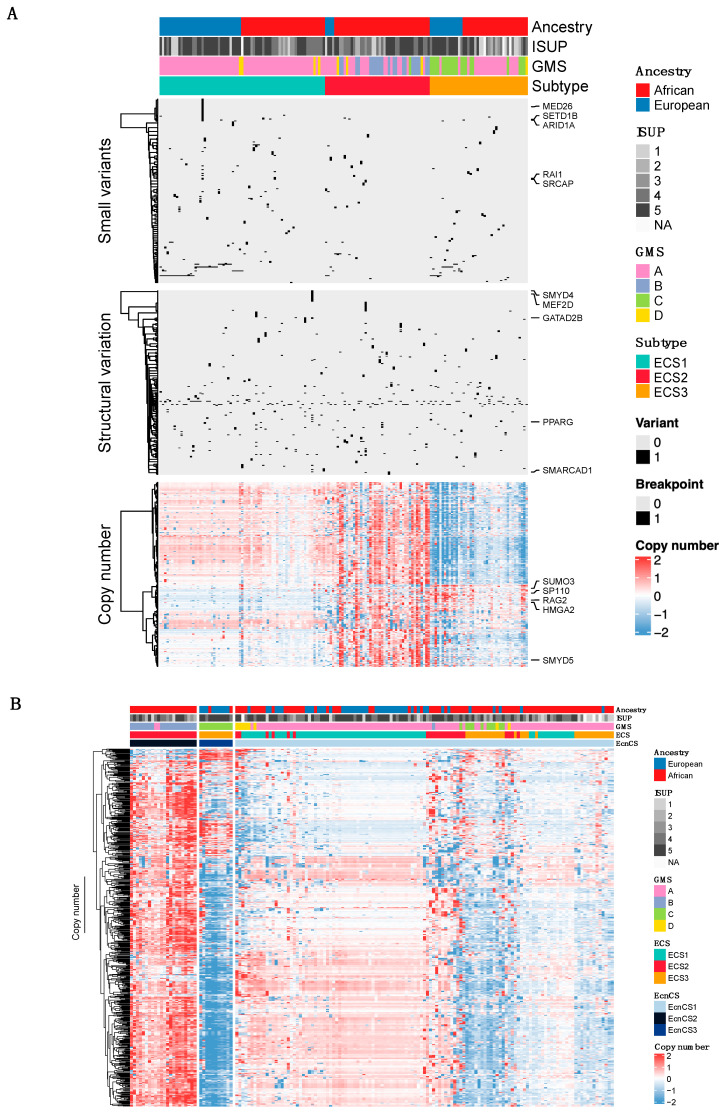
(**A**) Consensus clustering heatmap, based on 10 multi-omics integrative clustering algorithms, for somatic data (small variants, structural variation, and copy number alterations) spanning epigenetic machinery genes in 105 African- and 53 European-derived prostate tumors. For each variant data, the top five features are listed. Feature selection identifies complex cross-talk between different variant data, which may allude to biological significance driving cancer heterogeneity. (**B**) Hierarchical clustering heatmap, based only on somatic copy number alteration data spanning epigenetic machinery genes, for 105 African- and 53 European-derived prostate tumors. ECS, epigenetic cancer subtype; EcnCS, epigenetic copy number cancer subtype; GMS, global mutational subtype; ISUP, International Society of Urologic Pathologists; NA, not available.

**Figure 3 cancers-15-03462-f003:**
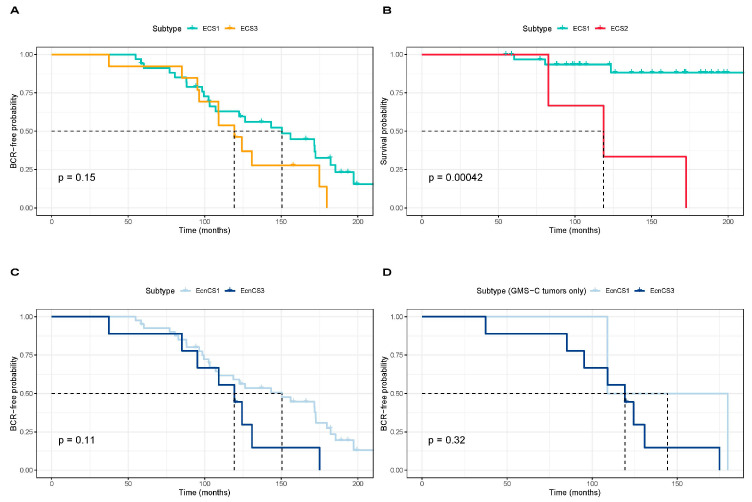
Kaplan–Meier curves of consensus clustering results for European patients. The probability estimates, 95% confidence intervals, and *p*-values (log-rank test) are indicated. (**A**) Kaplan–Meier curve of biochemical relapse (BCR)-free probability for ECS1 (*n* = 34) and ECS3 (*n* = 13) tumors. (**B**) Kaplan–Meier curve of the cancer survival probability for ECS1 (*n* = 34) and ECS2 (*n* = 3) tumors. (**C**) Kaplan–Meier curve of BCR-free probability for EcnCS1 (*n* = 41) and EcnCS3 (*n* = 9) tumors. (**D**) Kaplan–Meier curve of BCR-free probability for EcnCS1 (*n* = 2) and EcnCS3 (*n* = 9) tumors allocated to GMS-C. BCR, biochemical relapse; ECS, epigenetic cancer subtype; EcnCS, epigenetic copy number cancer subtype; GMS, global mutational subtype.

**Figure 4 cancers-15-03462-f004:**
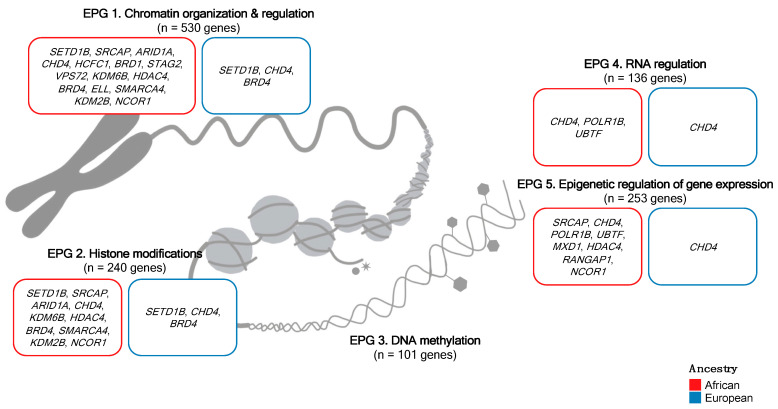
Top epigenetic machinery genes in African and European-derived tumors, somatically altered per epigenetic process group, that may be instrumental in epigenetic dysregulation and consequent prostate cancer oncogenesis. The total number of genes in each epigenetic process group is also displayed. Top genes were selected based on iClusterBayes feature selection, with a posterior probability > 0.5, as well as the presence of potentially damaging variants based on functional impact prediction and/or recurrence. No top genes were identified for epigenetic process group 3. EPG, epigenetic process group.

**Table 1 cancers-15-03462-t001:** Damaging variant summary for Africans and Europeans for each epigenetic process group, based on functional impact prediction.

		Africans (*n* = 109)		Europeans (*n* = 56)		
	Total Genes	Total Damaging Variants	Samples that Contain Damaging Variants	Total Damaging Variants	Samples that Contain Damaging Variants	*p*-Value *
EPG 1	530	71	37	26	22	0.4988
EPG 2	240	35	21	13	12	0.8375
EPG 3	101	5	5	3	3	1
EPG 4	136	12	9	5	4	1
EPG 5	253	31	23	15	13	0.8426

* Fisher’s exact test to compare variance in number of samples containing damaging variants between Africans and Europeans.

**Table 2 cancers-15-03462-t002:** Genes in Africans and Europeans containing damaging variants, based on functional impact prediction for each epigenetic process group.

	Africans	Africans and Europeans (Shared)	Europeans
	(Unique)		(Unique)
EPG 1	*AIRE*, *ARID1A* ^2^, *ARID1B* ^2^, *BAZ1B*, *BRD1*, *BRD7*, *BRWD1* ^2^, *CARM1*, *CHD1* ^1,2^, *CHD3* ^2^, *CXXC1*, *DNMT1*, *EHMT1*, *EHMT2*, *ELL*, *GLI1*, *HCFC1*, *HDAC4* ^1,2^, *HJURP*, *KAT8*, *KDM2B*, *KDM6B* ^1^, *KMT2D* ^2^, *MBD5*, *MED12*, *MEN1*, *NCOR1*, *NUP188*, *NUP93*, *PAX3*, *PELP1*, *POLR1A*, *POLR3A*, *PRDM16*, *PRMT6*, *PSIP1* ^2^, *RBBP5*, *REST*, *RNF20*, *RTF1*, *SATB2*, *SETD2* ^2^, *SMARCA1* ^2^, *SMARCA4* ^1^, *SRCAP*, *SSRP1*, *STAG2* ^1,2^, *TADA2B* ^1,2^, *TAF1*, *TAF6L*, *TPR*, *VPS72*, *WAPL*, *WHSC1L1* ^1^	*ATRX* ^2^, *BRD4*, *CHD4*, *DNMT3B*, *NCOR2* ^2^, *PADI1*, *PBRM1* ^1^, *SETD1B* ^2^	*AR*, *CBX2*, *CLOCK*, *ELOA*, *ERCC3*, *HNRNPA2B1*, *JAK2*, *KAT2B*, *KAT6A*, *KDM1B*, *KDM6A*, *KMT2B* ^3^, *KMT5C*, *PRDM5*, *RANBP2*, *RB1*, *RNF40*, *SIRT3*
EPG 2	*AIRE*, *ARID1A* ^2^, *ARID1B* ^2^, *CARM1* ^2^, *CHD3* ^2^, *DNMT1*, *EHMT1*, *EHMT2* ^2^, *HDAC4* ^1,2^, *KDM2B*, *KDM6B* ^1,2^, *KMT2D* ^2^, *MBD5*, *NCOR1*, *PRDM16* ^2^, *PRMT6*, *RBBP5* ^2^, *REST* ^2^, *SETBP1*, *SETD2* ^2^, *SMARCA4* ^1,2^, *SRCAP*, *WHSC1L1* ^1^	*BRD4*, *CHD4*, *DNMT3B*, *NCOR2* ^2^, *PBRM1* ^1^, *SETD1B* ^2^	*JAK2*, *KDM1B*, *KDM6A* ^4^, *KMT2B* ^3^, *KMT5C*, *PRDM5*, *SETD5*
EPG 3	*DNMT1*, *MBD5*	*DNMT3B*, *TDG*	*CBX2*
EPG 4	*ALG13*, *BAZ1B*, *CHD3* ^2^, *DNMT1*, *EHMT2*, *GSK3B*, *POLR1A*, *POLR1B*, *TDRD7*, *UBTF*	*CHD4*, *DNMT3B*	*ERCC3*, *KAT2B*, *SF3B1*
EPG 5	*AIRE*, *ALG13*, *BAZ1B*, *CHD3* ^2^, *DNMT1*, *EHMT2*, *FOXO3*, *GSK3B*, *HDAC4* ^1,2^, *MXD1*, *NCOR1*, *POLR1A*, *POLR1B*, *RANGAP1*, *REST*, *SRCAP*, *UBTF*, *XRCC6*	*CHD4*, *DNMT3B*, *NCOR2*, *TDG*, *TP53* ^1,2,3,4^	*ERCC3*, *HDAC9*, *KAT2B*, *RANBP2*, *RXRB*, *SF3B1*, *SIRT3*, *STAT3*

EPG, epigenetic process group. ^1^ Genes that contain more than one damaging variant in an EPG (Africans). ^2^ Genes identified as potentially damaging in an EPG, based on recurrent somatic variant identification (Africans). ^3^ Genes that contain more than one damaging variant in an EPG (Europeans). ^4^ Genes identified as potentially damaging in an EPG, based on recurrent somatic variant identification (Europeans).

**Table 3 cancers-15-03462-t003:** Genes in Africans and Europeans identified as potentially damaging for each epigenetic process group, based on recurrent somatic variants.

	Africans	Africans and Europeans	Europeans
	(Unique)	(Shared)	(Unique)
EPG 1	*ARID1A* ^1,2^, *ARID1B* ^1,2^, *ATRX* ^1,2^, *BRWD1* ^1,2^, *CHD1* ^2^, *CHD3* ^1,2^, *CHD7*, *DPF3*, *ELP2*, *EP300* ^1^, *GATAD2B*, *GLI3*, *HDAC1*, *HDAC3* ^1^, *HDAC4* ^2^, *KMT2A*, *KMT2D* ^1,2^, *NCOR2* ^1,2^, *NUP35*, *PSIP1* ^1,2^, *SETD1B* ^2^, *SETD2* ^1,2^, *SMARCA1* ^1,2^, *STAG2* ^1,2^, *TADA2B* ^2^, *XPO1* ^1^	*KMT2C* ^1^	—
EPG 2	*ARID1A* ^1,2^, *ARID1B* ^1,2^, *ARID5B* ^1^, *BRMS1*, *CARM1* ^2^, *CHD3* ^1,2^, *EHMT2* ^2^, *EP300* ^1^, *GATAD2B*, *HDAC1*, *HDAC3* ^1^, *HDAC4* ^2^, *KDM6B* ^2^, *KMT2A*, *KMT2D* ^1,2^, *NCOR2* ^1,2^, *PRDM16* ^2^, *RBBP5* ^2^, *REST* ^2^, *SETD1B* ^2^, *SETD2* ^1,2^, *SMARCA4* ^1,2^	*KDM6A* ^1,3^, *KMT2C* ^1^	—
EPG 3	—	—	—
EPG 4	*CHD3* ^1,2^, *HDAC1*	—	—
EPG 5	*CHD3*^1,2^, *HDAC1*, *HDAC4* ^2^	*TP53* ^1,2,3^	—

EPG, epigenetic process group. ^1^ Pan Cancer Analysis of Whole Genomes candidate cancer mutational driver. ^2^ Contains damaging variants in Africans, based on functional impact prediction. ^3^ Contains damaging variants in Europeans, based on functional impact prediction.

## Data Availability

Data used in this study were published by Jaratlerdsiri et al., 2022, and made accessible via the European Genome-Phenome Archive (EGA; https://ega-archive.org, accessed on 1 June 2022) under study accession EGAS00001006425 and dataset accession EGAD00001009067 (Southern African Prostate Cancer Study, SAPCS) and EGAD00001009066 (Garvan/St. Vincent’s Prostate Cancer Study).

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
