# Peer review of "Alterations in the Epigenetic Machinery Associated with Prostate Cancer Health Disparities"

_cancers, 2023, doi:10.3390/cancers15133462_

Round 1
Reviewer 1 Report
The authors in this work studied the alterations in the epigenetic machinery associated with prostate cancer health disparities. The work tried to address an important issue in prostate cancer research. The experiments were well designed, and the work was conducted nicely. The manuscript is well writing.
The authors hypothesized that dysregulation among the roughly 656 epigenetic genes may contribute to prostate cancer health disparities. What is the basis to make this hypothesis? The authors should explain a little more on how these 656 genes were selected.
Author Response
The following sentence was added to the introduction to clarify the selection of the 656 genes and reads as follows: "Consultation of the PathCards database [31] and a review of the literature [1,7,8,32–34] identified 656 genes (in)directly related to all known epigenetic processes, several of which have been implicated previously in PCa."
Reviewer 2 Report
In the manuscript entitled “Alterations in the epigenetic machinery associated with prostate cancer health disparities” authors have identified a number of genetic variants and copy number alterations associated with African and European ancestry by analyzing the prostate tumor genomic data of 109 men of Southern African and 56 men of European Australian ancestry. Using sequencing and bioinformatic analysis authors have shown interesting findings and novel epigenetic drivers of prostate cancer in patients with African ancestry.
Comments:
# Please mention the basis of selection criteria of 656 gene, called epigenetic genes. Were all the genes known to be involved in epigenetic modification process?
# What authors mean when referring to damaging variants?
# It would be interesting to include the type of epigenetic modification present in the ancestry based on genetic alterations identified in these groups.
Author Response
Please mention the basis of selection criteria of 656 gene, called epigenetic genes. Were all the genes known to be involved in epigenetic modification process?
Response: For further clarification the following sentence was added to the introduction: "Consultation of the PathCards database [31] and a review of the literature [1,7,8,32–34] identified 656 genes (in)directly related to all known epigenetic processes, several of which have been implicated previously in PCa."
What authors mean when referring to damaging variants?
Response: Damaging variant definition was mentioned in the Methods section. See the sentence: "A variant was considered to be potentially damaging if identified by SIFT as “Deleterious” or “Deleterious - Low Confidence”, or if identified by PolyPhen as “Possibly Damaging” or “Probably Damaging”."
It would be interesting to include the type of epigenetic modification present in the ancestry based on genetic alterations identified in these groups.
Response: Future studies by this team will interrogate for epigenetic modifications. Studies are underway.